# A Review of Topical and Systemic Vitamin Supplementation in Ocular Surface Diseases

**DOI:** 10.3390/nu13061998

**Published:** 2021-06-10

**Authors:** Paolo Fogagnolo, Stefano De Cilla’, Micol Alkabes, Pierfilippo Sabella, Luca Rossetti

**Affiliations:** 1Ophthalmology Unit, Department of Health Sciences, San Paolo Hospital, Università degli Studi di Milano, 20142 Milan, Italy; pierfilipposabella@gmail.com (P.S.); luca.rossetti@unimi.it (L.R.); 2Ophthalmology Unit, Ospedale Maggiore della Carita, 28100 Novara, Italy; stefano.decilla@med.uniupo.it (S.D.C.); micol.alkabes@med.uniupo.it (M.A.)

**Keywords:** ocular surface disease, dry eye disease, goblet cells, vitamin A, vitamin D, Sjogren syndrome, diabetes, neuropathic ocular pain, glaucoma

## Abstract

In the homeostasis of the ocular surface, vitamins play a critical role in regulating inflammatory responses and promoting cell differentiation, development and correct function. Systemic vitamin supplementation has been available for many decades; in recent years, thanks to pharmacological advancements, topical vitamin delivery has also become available in an attempt to better treat ocular surface disease (OSD) and dry eye disease (DED). In this paper, we reviewed the current evidence on the role of vitamin supplementation in OSD and DED. We originally searched the PubMed archive, inspected the references and restricted the search to pertinent papers. The body of evidence was evaluated using the amelioration of both signs and symptoms as the outcome, when available. We found that in patients with vitamin deficiency, systemic supplementation of Vitamin A is effective in treating OSD, reducing both DED signs and symptoms. Additionally, systemic supplementation of vitamin D is useful in reducing DED symptoms and increasing tear volume. Vitamin A is also effective in reducing DED signs and symptoms when administered locally. The efficacy of supplementation with other vitamins is still not fully proven. In conclusion, the inclusion of vitamins into the treatment strategies for OSD and DED allows for better treatment customization and better outcomes in these patients.

## 1. Introduction

Ocular surface disease (OSD) is an inclusive term encompassing the abnormalities of the ocular surface (OS) as a functional unit, including multiple epitheliopathies, inflammation, surface abnormalities and distortions, which all affect tear dynamics. It also includes dry eye disease (DED), a multifactorial disease characterized by loss of homeostasis of the tear film, OS inflammation and damage, and neurosensory abnormalities. In DED, these changes are accompanied by increased tear evaporation and hyperosmolarity, and typical symptoms [1]. DED may also be caused by dysregulation of the immune system, such as Sjogren syndrome (SS).

In the homeostasis of the OS, vitamins play a critical role. Vitamins A, C, and E are physiologically present in the OS and are essential for cell differentiation, development, and correct function [2]. Vitamin D and B have a key regulation activity of the immune and nervous systems, respectively [2]. Vitamin deficiencies are frequently associated with both OSD and DED, and systemic supplementation has proved to be beneficial in subgroups of patients with low vitamin intake.

The treatment of DED is stepwise and aims at controlling its risk factors and specific pathophysiological patterns. Lubrication is a mainstay of DED treatment at all stages [3]. Recently, many lubricating eyedrops have been enriched with compounds (amino acids, sugars, vitamins) in an attempt to promote OS homeostasis and control local inflammation. However, the usefulness of topical and systemic vitamin supplementation in DED is still controversial.

The objective of this review is to address whether topical and systemic vitamin supplementation are useful in improving signs and symptoms in patients with DED and, more in general, OSD.

## 2. Materials and Methods

This review was performed at the Department of Health Sciences, Università degli Studi di Milano, San Paolo Hospital, Milan, Italy. We originally searched the PubMed archive on 01 March 2021 with the keywords “vitamin” AND “ocular surface disease” OR “dry eye disease”. We inspected the results of 4230 papers and identified 254 papers, providing the following useful information: (1) preclinical evidence of efficacy; (2) clinical evidence of efficacy. Two evaluators independently inspected the full text and references to retrieve relevant papers. Excluded were reports with duplicated results, case series with 10 patients or less, papers with no abstract or papers not written in English. No restrictions on the presence of a control group or length of follow-up were applied. In the end, 55 studies were analyzed in this review.

## 3. Results

### 3.1. Vitamin A

#### 3.1.1. Preclinical Evidence

There is large preclinical evidence supporting the beneficial effects of vitamin A administration (either systemically or topically) on the ocular surface. The mechanisms of action of vitamin A on the OS are reported in Figure 1. Vitamin A increases conjunctival mucin expression and promotes correct corneal and conjunctival wound healing, reducing keratinization. These effects have been shown by several studies using different animal models of DED: vitamin A promotes wound healing [4] by directly improving epithelial repair [5,6], and reducing cell apoptosis [7] and corneal keratinization [8]. These effects are mediated by an increase in tear volume [9,10] and quality [5]. On a rabbit model of OSD, due to antiglaucoma treatments, topical vitamin A was as effective as topical cyclosporine in reducing OS inflammation, as evaluated by impression cytology [11].

The mechanism of action of vitamin A is complex. It upregulates the cytochrome P450 synthesis of eicosanoids in in vitro models of human conjunctival and cornea cells [12]. Eicosanoids exhibit potent inflammatory and angiogenic properties, and upregulate secretory phospholipase A (2) group IIA genes, which are responsible for increased mucin 16 expression [13]. Expression of mucin 16 in the cornea glycocalyx provides a protective barrier for the epithelium [13]. Vitamin A also stimulates the synthesis of mucin 4 (physiologically absent on human OS), but not directly the synthesis of mucin 1 (one of the most expressed mucin on the OS secreted by goblet cells) [14]. Finally, vitamin A also down-regulates androgen receptor expression on the ocular surface [15].

#### 3.1.2. Clinical Studies on Systemic Supplementation

Vitamin A deficiency is a public health problem in about 60 countries worldwide, causing increased mortality and morbidity in affected children [16]. Vitamin A deficiency causes OS changes consisting of progressive goblet cell loss up to complete absence [17], severe DED, and corneal punctate keratopathy [18]. Severe involvement causes epithelial metaplasia and keratinization, potentially leading to xerophthalmia (consisting of OS keratinization with Bitot spots) and keratomalacia. There is large evidence from population studies that systemic vitamin A supplementation reduces OS changes [19,20] at any stage of the disease by guaranteeing goblet cell repopulation of the conjunctiva and an increase in their density [21], corneal re-epithelization [21,22], and increased tear quality [23]. Oral and intramuscular vitamin A supplementation achieved similar effects [24].

#### 3.1.3. Clinical Studies on Topical Supplementation

Due to OS changes occurring in vitamin A deficiency, topical vitamin A formulations have also been introduced in the clinical scenario; recent technological advances have also made it possible to solubilize it into eye drops. The effects of topical administration were evaluated by eight papers, whose main data are reported on Table 1.

The first published paper was non-controlled and showed that vitamin A ointment could ameliorate symptoms and OS signs on all of the 22 severe OSD patients unresponsive to previous treatments included in the study [32].

The following seven papers were controlled versus placebo (one study) [31], other lubricating ointments or gel (four) [26,27,29,30], cyclosporine A [27] and moist chamber [25]. It should be kept in mind how placebo may have an effect as well in the OS, compressing the dynamic range for clinical differentiation between active treatment and placebo [33]; in fact, a positive effect comparing baseline to final data was also shown in two placebo arms [27,31]. All of the studies confirmed that vitamin A was superior to the control in ameliorating OS parameters, except that by Gilbard et al., whose setting is hardly replicable because of the limited description of the study population, design and methods.

When severe DED patients were treated with vitamin A, the effects on both symptoms and signs were largely demonstrated. The parameters most positively affected were symptoms (in particular, blurred vision), goblet cell density, epithelial staining, Schirmer and tBUT. An interesting paper compared vitamin A to cyclosporine A, and both treatments were largely effective in reducing DED signs and symptoms, also promoting goblet cell proliferation by more than 70% over a 3-month follow-up [28]. Vitamin A achieved higher performances than cyclosporine in terms of recovery of blurred vision and Schirmer test, and less discontinuation due to side effects. In glaucoma patients, vitamin A is superior to carbomer gel in improving the damage caused by chronic treatments to goblet cells: a significant increase, superior to 50% from baseline, was found over a 6-month interval using both confocal microscopy and impression cytology; both treatments were effective in reducing symptoms in these patients [26]. Soong et al. confirmed that vitamin A reduces epithelial keratinization; the failure to show a change in symptoms and Schirmer test is expected on a population also including severe OSD other than DED [30].

From the data available in the literature, systemic supplementation of vitamin A would be necessary in patients with insufficient intake; topical supplementation could be useful in patients with severe DED in whom previous treatments were unsatisfactory. Efficacy was particularly shown in patients with low tear quality due to mucin impairment and lower ferning test scores [26,29], including poorly responsive glaucoma patients [26], as also confirmed by recent publications by our group [34,35,36] on a multivitamin formulation including vitamin A (details are given in Section 3.4.2).

### 3.2. Vitamin B

#### 3.2.1. Preclinical Evidence

Dexpanthenol is a provitamin and an alcohol analog of vitamin B5 (pantothenic acid). It has a well-known effect in promoting epithelial healing. The effects on the OS were recently confirmed by Mencucci et al. in an in vitro study of human conjunctival and corneal epithelial cells treated for 18 h with a solution containing dexpanthenol 5% and polyvinyl alcohol [37]. Additionally, the role of vitamin B12 in the maintenance of a correct nerve trophism has been largely demonstrated in preclinical studies [2].

#### 3.2.2. Clinical Studies on Systemic Supplementation

Different studies showed that patients with SS are frequently affected by vitamin B deficiency, in particular B12, B6, B1 and B2. Vitamin B12 deficiency is associated with SS in 10–25% of patients [38,39]; it may also be a source of neuropathic ocular pain [40]. The efficacy of oral supplementation with vitamins of group B is debated. In spite of the evidence of a preexisting deficiency, a 3-month oral integration with vitamin B6 in patients with SS did not modify the expression of interleukin 2. The authors explained their results, reporting that the deficiency was not severe enough to affect the immune system [41]. Unfortunately, no clinical parameters were measured in these patients. Additionally, McKendry reported a lack of significant advantage when SS patients are supplemented with vitamin B6, C and linolenic acid [42], whereas Horrobin reported an improvement in tear secretion thanks to the use of fatty acids, vitamin B6 and C [43].

Nebbioso et al. performed a placebo-controlled study of the effects of a food supplement containing vitamin B1 and B2, forskolin and rutin on 38 glaucomatous patients with OSD due to chronic exposure to preservatives. The treatment was effective in improving both DED signs and symptoms. Schirmer test scores improved from 4.4 ± 2 s at baseline to 9.7 ± 2.9 s after one month, tBUT from 4.5 ± 2.7 to 7.9 ± 2.1 s, and ocular surface disease index (OSDI) score from 38.7 ± 14.5 to 22.5 ± 8.7 [44].

Finally, supplementation with vitamin B12 for 3 months on a group of 45 patients with vitamin B12 deficiency and neuropathic ocular pain was able to nearly halve OSDI score (from a mean of 67 at baseline to 37 at the end of the study) and to increase both tear secretion (mean ST changed from 2.8 to 15.0 s) and stability (mean tBUT changed from 3.5 to 11.7 s). Of note, these improvements were superior to those occurring in a group of severe DED treated with lubricating eye drops and cyclosporine [40].

#### 3.2.3. Clinical Studies on Topical Supplementation

The effects of topical treatment with vitamin B have been the object of three recent papers; all of them evaluated vitamin B12. One study evaluated the advantage of a nebulization of the OS with vitamin B12 twice weekly for 3 months in patients with DED [45]. Treatment ameliorated both symptoms and signs: mean OSDI was 47 at baseline and 29 at the end of the study; tBUT ameliorated from 3.0 to 5.5 s; mean corneal staining grade was 0.7 at baseline and 0 at the end of the study. Significant amelioration under confocal microscopy was also shown: increased epithelial density, increased sub-basal plexus density, and decreased number of dendritic cells. An increase in sub-basal plexus density and corneal esthesiometry was also shown in an 18-month placebo-controlled perspective study on patients with diabetes and DED receiving an eye drop of vitamin B12 and citicoline, and these changes were again associated with an improved OS milieu [46]. Obviously, the effects of vitamin B12 alone cannot be drawn by these data. Improvements in DED signs and symptoms were also shown in DED patients using an eye drop containing vitamin B12 and hyaluronic acid four times daily for one month, but again, the effect of vitamin B12 alone could not be isolated [47].

### 3.3. Vitamin D

#### 3.3.1. Preclinical Evidence

Vitamin D concentration in human tears is higher than serum [48]. It is dosable in aqueous and in vitreous fluids, and a specific receptor for vitamin D is present in corneal epithelial cells and stroma, as well as in the lens, scleral fibroblasts and epithelial cells of the retina and ciliary body [49]. In the human cornea, there is an active metabolism of vitamin D thanks to the presence of the enzyme 1-alpha-hydroxylase, which activates vitamin D3 to its active metabolite calcitriol [49].

Vitamin D plays a key role in modulating immune system activity and inflammation, which are both critical in maintaining OS homeostasis. Topical treatment with vitamin D is associated with better antibacterial response of the OS [50]. The local concentration of calcitriol is inversely associated with inflammatory levels in DED models, so that it has been postulated that vitamin D may *per se* be a treatment for DED [51]. The anti-inflammatory effects of vitamin D were shown in animal models of neovascularization [52], exposure to preservative toxic damage [53] or hyperosmolarity agents [54], and cauterization of the orifices of meibomian glands [55].

Studies on knockdown mice showed that vitamin D is also essential for the function of limbal stem cells and epithelium [56], in part through regulation of the release of growth factors and of metalloproteinases [57]. Vitamin D also regulates the barrier function of corneal epithelium. It in fact modulates the retinoid X receptor, which in turn controls cell differentiation and ensures maintenance of tight junctions [58].

#### 3.3.2. Clinical Studies on Systemic Supplementation

The impact of vitamin D on DED has been the object of several studies published over the last few decades, frequently with controversial conclusions. Recently, the results of these studies were summarized in a meta-analysis confirming that vitamin D is likely a risk factor for DED, as patients with DED have a serum vitamin D level lower than in controls by 4 ng/mL (95% CI −6.58; −1.40) [59]. Vitamin D deficiency is associated with worse DED symptoms (OSDI score is higher than in controls by 11 points) and less tear secretion (ST is lower by a mean of 6 mm/5 min); tear stability (tBUT) is apparently similar in patients with and without vitamin D deficiency [60]. Low vitamin D levels would hypermodulate nociception status [61,62].

Vitamin D supplementation may be effective, particularly in DED patients refractory to conventional treatments and with vitamin D deficiency [63]. Yet, the impact of oral supplementation in preventing DED or ameliorating DED signs and symptoms still has to be convincingly proven by prospective studies [59].

#### 3.3.3. Clinical Studies on Topical Supplementation

No studies are available on the effects of supplementation with topical vitamin D alone. Results of topical multivitamin supplementation including vitamin D are given in Section 3.4.2.

### 3.4. Vitamin C, E, or Combined Vitamins

#### 3.4.1. Preclinical Evidence

No relevant papers are available on the effects of supplementation with vitamin C and E, or multivitamins in models of OSD or DED.

#### 3.4.2. Clinical Studies on Systemic Supplementation

Vitamin C and E have been orally administered together in acute severe OSD diseases such as chemical injuries [63], or in patients with locally chronic increase of oxidative stress such as diabetic patients [64,65]. Three papers by two groups of authors investigated the changes occurring in OS parameters in diabetic patients after oral treatment with vitamin C and E [64,65,66]. The studies were not placebo-controlled; the treatment period was 10 and 30 days. All studies reported a significant improvement in OS parameters; the median increase was about 2.5 s for tBUT and 3.0 mm for Schirmer [64,65]. One study also evaluated tear nitrite concentration, ferning test and goblet cell density [64]: the authors concluded that the amelioration of the OS induced by vitamin C and E supplementation can be explained by a reduction in oxidative damage produced by nitric oxide and other free radicals, which positively reflects on the mucin secretion of goblet cells.

#### 3.4.3. Clinical Studies on Topical Supplementation

No studies exploring the effects of topical vitamin C administration are currently available.

Two formulations of vitamin E and coenzyme Q10 have been evaluated in four different studies on different groups of DED patients (patients receiving cataract surgery, women in menopause receiving antidepressants, subjects exposed to local irritants, children) [67,68,69,70]. All the studies showed that these treatments are effective in ameliorating OS homeostasis, tear stability [67,68,69,70] and in restoring the normal corneal innervation under confocal microscopy [67]. Unfortunately, it is not possible to isolate the effects of vitamin E alone from these data.

A topical multivitamin approach has recently become available thanks to pharmacological advances. We recently tested the efficacy of a novel lubricating eye drop containing vitamin A and D and Omega 3 on a liposomal nano-dispersion on different types of DED [34] including hyper-evaporative DED. The treatment was strongly effective in increasing tear stability, reducing tear evaporation, reducing OS inflammation and promoting reepithelization compared with baseline [34,35]; the results were not inferior to a competitor lipid eye drop [34]. A subgroup of glaucoma patients with pathological ferning test was evaluated after 6 weeks of treatment with vitamin A, and normal ferning patterns were found in 75% of cases [36].

This formulation was tested on post-surgical DED as a model of iatrogenic damage of corneal structure and sub-basal nerve plexus [35]. Patients were randomized with a 1:1 ratio to simple observation or to receive vitamin A and D and Omega 3 on a liposomal nano-dispersion from two weeks before cataract surgery to two weeks after. We showed that the two groups had similar baseline characteristics, and that study treatment was helpful in preserving stable tBUT throughout the study, whereas a progressive reduction was shown in the observation group; epithelial staining was also lower in the treated arm. Our clinical observation supports the role of vitamin A and D and omega 3 in improving signs and symptoms in patients with iatrogenic DED due to cataract surgery [34], although the efficacy of the single components of the eye drop could not be ruled out.

### 3.5. Safety of Vitamin Supplementation

Vitamin A has been administered systemically and locally for decades; therefore, its safety profile is largely documented. The most severe adverse effect of systemic retinoids abuse is teratogenicity [71]. The most common acute adverse effect of topical retinoids is blepharoconjunctivitis [30,72], with skin irritation and peeling, and conjunctival hyperemia [71]. These local side effects were apparently not dose-related; interestingly, another trial found no local side effects over a 28-day observation [26]. The major concern of chronic long-term topical treatment is its detrimental effect on meibomian glands, potentially resulting in progressive atrophy of acini and hyposecretion of oils. This effect is reversible on discontinuation of the drug [72].

The ocular safety profile of other vitamins is apparently high. No significant side effects were reported for topical vitamin B, D, and E supplementation, apart from occasional eye burning in patients receiving a combination of vitamin E and coenzyme Q10 [67].

## 4. Discussion

This paper aimed at reviewing the evidence on the efficacy of vitamin supplementation to prevent DED and other OSD.

There is large preclinical evidence that vitamin deficiencies are associated with abnormal cell metabolism, potentially leading to cell degeneration or loss. In the OS, vitamin A, C, and E deficiencies firstly affect goblet cells (the smallest structures of the OS with no mitotic activity) and secondarily, also epithelial cells and meibomian glands [17,21]. These changes have been clinically demonstrated in population studies on patients suffering from vitamin A deficiency, which is still, today, a sanitary emergency in underdeveloped areas [18,19,20]. Additionally, a low plasma level of vitamin D is frequently associated with DED [59], whereas deficiencies of vitamin B, C and E are less common nowadays.

The question of whether vitamin supplementation is capable of recovering DED or OSD is more challenging. For vitamin A, mass treatment has been shown to be effective in halting epithelial metaplasia and keratinization, and this beneficial effect was also present at early stages of the disease, allowing for the normalization of goblet cell density [21,23]. The duration of such effects has not been explored yet: no prospective studies are available correcting the results for long-term micronutrient plasma level and dietary intake modifications in patients receiving mass treatment. Vitamin D supplementation was effective in DED patients with vitamin D deficiency, but prospective studies on the course of the disease are required to correctly measure the effects in these patients [59]. Even less evidence is available for the supplementation of other vitamin deficiencies. As a general rule, clinicians should be more aware of the relevance of systemic vitamin deficiency for OS homeostasis. Serum vitamin levels should be checked in OSD and DED patients; in case of vitamin deficiency, systemic integration should be considered in order to ameliorate whole body homeostasis and treat any subclinical or undetected manifestation of avitaminosis. Yet, chronic systemic supplementation may cause suboptimal adherence, especially for patients on multiple therapies or those concerned by the high cost of medications.

Local vitamin supplementation might be an appropriate choice when specific local damage is shown (for example, patients chronically treated with preserved or proinflammatory medications) because it has the advantage that it could be tailored to the patient on the basis of specific OS findings. Topical vitamins are, in most cases, combined with lubricating molecules; adherence would not be a major issue as DED patients are, in general, strongly symptomatic and they regularly use lubricating eye drop supplementation. In countries where lubricating drops are not reimbursed, the addition of topical vitamins is associated with a limited increase in costs for the patients, but individual costs may be high due to patent protection in countries where other lubricating eye drops are reimbursed.

To date, among local supplementations, only vitamin A gained a sufficient proof of efficacy. This product has been used for decades in patients with severe DED poorly responsive to other treatments, but it is also effective in initial stages of the disease thanks to its ability to improve mucin layer and corneal wettability. In all of these studies, vitamin A supplementation ameliorated the OS compared with baseline [25,26,27,28,29,30,31,32], and in all except one [27], vitamin A was superior to the vehicle (the vehicle may have a positive effect in DED, compressing the dynamic range for clinical differentiation between active treatment and placebo [33]). Of note, a recent study showed that topical vitamin A would be as effective as cyclosporine in DED patients not responding to standard treatments [26].

We recently showed that a combination of vitamin A, D and Omega 3 is effective in evaporative forms of DED, in patients with post-surgical DED and in those with glaucoma and mucin impairment. Additionally, studies on vitamin B12 confirmed that it may be useful to promote nerve activity, whereas vitamin D could be added in cases of significant local inflammation or immune system dysregulation (provided that vitamin D deficiency is not proven; in that case, systemic supplementation would be indicated). A recent case report also suggested that local vitamin D could be useful to treat obstructive meibomian gland dysfunction and hyperkeratotic changes of the eyelids [73]. Yet, the proofs of efficacy and the safety profile of local supplementation of vitamins other than A require further verification for several reasons: these treatments have been recently introduced and prospective studies are not available or are limited in number; frequently microelements are added to other compounds and their specific activity cannot be measured precisely; frequently, patients are highly selected, and follow-up is limited in time; finally, these studies do not correlate local findings with vitamin plasma level and dietary intake.

The limits of the studies currently available should be considered in order to plan better designed studies in the next future, and they should not discourage clinicians from seeking more individualized treatments for our patients. OSD is a multifactorial disease encompassing very different scenarios, and the identification and treatment of systemic vitamin deficiencies, as well as the integration of micronutrients into lubricating eye drops, allow for better tailored treatments, and possibly better adherence and higher efficacy. A few examples of treatment customization are reported in Table 2.

In turn, assuming that patients with OSD should be treated in the same way as they share common signs and symptoms is an unsuccessful strategy in most cases, because it does not consider the very different pathogeneses underlying these common findings.

## 5. Conclusions

OSD and DED are complex, multifactorial diseases whose management needs a careful assessment of patients, both systemically and locally. In this review, we provided evidence that plasma vitamin levels should be measured and, in case of deficiencies, systemic supplementation with vitamin A and D is useful to improve signs and symptoms in these patients. Additionally, local vitamin A supplementation is useful to improve goblet cell density and epithelial health. Other vitamin supplementation might be useful in specific conditions, but it is still necessary to obtain more evidence from appropriately designed studies.

## Figures and Tables

**Figure 1 nutrients-13-01998-f001:**
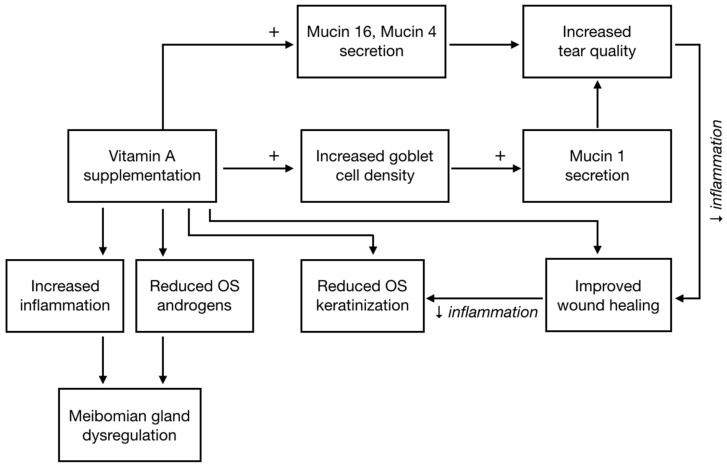
Effects of vitamin A supplementation on the ocular surface.

**Table 1 nutrients-13-01998-t001:** Main features and outcomes of the studies on topical supplementation on vitamin A.

Author, Year	Patients	Follow-Up	Active Arm	Control	Outcomes	Results
Babamohamadi, 2018 [25]	38 unconscious patients	5 days	VA ointment	Moist chamber	ST, OSS	VA achieved higher ST than moist chamber
Cui, 2016 [26]	30 glaucoma patients	6 months	VA 0.1% gel	Carbomer 0.2% gel	Symptoms, tBUT, confocal microscopy, impression cytology	Both treatments are effective vs. baseline in reducing symptoms. VA apparently superior in recovering goblet cells
Gilbard, 1989 [27]	11 with severe DED	5–22 weeks	VA ointment (0.1–0.01%)	mineral oil ointment	ST, OSS, tear film osmolarity	Similar effects in the two groups
Kim, 2009 [28]	150 with DED unresponsive to conventional treatments	3 months	VA ointment 0.05%	Cyclosporine A 0.05%	ST, tBUT, OSS, goblet cell density, impression cytology	VA ameliorated all study parameters, similarly to Cyclosporine. ST and blurred vision recovered faster in VA group. Discontinuation: 10% (VA), 14% (cyclosporine), 18% (control)
Selek, 2000 [29]	22 patients with severe DED	7 days	VA emulsion (0.01%)	Polyvinyl alcohol	Symptoms, ST, tBUT, OSS, Ferning	VA improved ST, tBUT, OSS and ferning vs. baseline. VA superior to placebo for ST and tBUT
Soong, 1988 [30]	116 with severe OSD or severe DED	4–8 months	VA ointment (0.01%)	Petrolatum ointment	Symptoms, keratinization (impression cytology), ST	VA improved conjunctival keratinization; symptoms and ST unchanged.
Toshida, 2017 [31]	66 DED patients	28 days	VA solution (0.05%)	Placebo eyedrop	Symptoms, OSS, tBUT, ST	VA improved symptoms and blurred vision; VA superior to placebo for OSS.
Tseng, 1985 [32]	22 with severe OSD and DED unresponsive to conventional treatments	NA	VA ointment (0.01–0.1%)	None	Symptoms, keratinization (impression cytology), OSS, ST, visual acuity	All parameters ameliorated

DED, dry-eye disease; NA, not available; OSS, ocular surface staining; ST, Schirmer test; tBUT, tear break-up time; VA, vitamin A.

**Table 2 nutrients-13-01998-t002:** How is it possible to customize treatment for DED and OSD today?

Clinical Question	Suggested Tests and Treatments
Patients with DED (Sjogren syndrome or not), poorly satisfied by topical treatments	Test plasma level of vitamin D. In case of deficiency, treat and monitor it
Patient with DED, chronically treated with anti-glaucomatous medications	Consider topical treatment with vitamin A to ameliorate the trophism of goblet cells and epithelial cells. Additionally, topical vitamin D may enhance barrier function
DED associated with significant inflammation	Test plasma level of vitamin D. In case of deficiency, treat and monitor it. Topical vitamin D supplementation may also be chosen
DED associated with meibomian gland dysfunction and hyperkeratotic changes of the eyelid	Test plasma level of vitamin D. In case of deficiency, treat and monitor it. Consider topical treatment with vitamin D to ameliorate eyelid abnormalities
Neuropathic ocular pain	Test plasma level of vitamin B and D. In case of deficiencies, treat and monitor them. Additionally, vitamin C and E supplementation may be effective. Check sub-basal nerve function (esthesiometry and/or confocal microscopy). Consider a topical/systemic treatment with vitamin B12

DED, dry-eye disease.

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
