# Peer review of "A Review of Topical and Systemic Vitamin Supplementation in Ocular Surface Diseases"

_nutrients, 2021, doi:10.3390/nu13061998_

Round 1

Reviewer 1 Report

  1. The manuscript needs professional English editing. Some grammer and spelling mistakes are noted.
  2. Line 80 , 81 ,You have to defined the abbreviation of MUC4 and MUCI .
  3. It is better to provide figures about the mechanism of Vitamin A on dry eye treatment . 
  4. Which subtype of dry eye will be benefited from vitamin treatment?Please add more information.
  5. The safety of systemic or topical vitamin treatment should be addressed.
  6. The evaluation of outcome between different studies need to be discussed.

Author Response

Reviewer 1

  1. The manuscript needs professional English editing. Some grammer and spelling mistakes are noted.

Done as recommended

  1. Line 80 , 81 ,You have to defined the abbreviation of MUC4 and MUCI .

Done as recommended

  1. It is better to provide figures about the mechanism of Vitamin A on dry eye treatment .

Done as recommended

  1. Which subtype of dry eye will be benefited from vitamin treatment?Please add more information.

This is an interesting observation. In most studies, a modern subdivision into hyposecretive or evaporative DED was not done (for many studies at the time of publication this classification had not been done yet). We highlighted available info whenever possible, and we discussed this issue.

  1. The safety of systemic or topical vitamin treatment should be addressed.

A section before Discussion has been added to address safety.

  1. The evaluation of outcome between different studies need to be discussed.

We tried to discuss this relevant topic more clearly in this revision

Reviewer 2 Report

I find the topic very interesting, so in that sense am positively inclined to see more work along these lines published, but that said while I keep trying to like this paper and it falls short still.  Trying to understand why I react that way to it, there seem to be three things:  

1) The authors survey 4230 papers and state they arrive at 254 providing pre-clinical and/or clinical evidence of efficacy, yet "in the end 55 studies were included in this review," which is approximately one out of five of those stipulated to meet their gross screening criteria.    No mention is made as to why the other four out of five were rejected.  Having explored these questions in some depth over 15 years myself, there are many studies of which I am aware which seem to have been dismissed.  Perhaps the authors had good reason to reject them, but we are given no clue as to why.  It is a very broad field of inquiry and I can't fault them for wanting to restrict their analysis to some digestible degree, but I would much rather they restrict their analysis to, say "topical vitamin impact" and offer more comprehensive exploration of their review criteria and findings than try to encompass the entire concept and leave me wondering why some studies were included and others not.  

2) Consideration of clinical studies on oral vitamin A, while perhaps needed to address the stated "systemic and topical" aspect of this paper, were not something which struck me as helpful (novel) contribution to the knowledge base.  Every ophthalmologist should receive textbook instruction regarding Vitamin A deficiency and xerophthalmia--nearly 100 years under study--and while perhaps the authors felt this needed due attention given the breadth of their topic, it was distracting to me.  The discussion of topical supplementation was much more interesting and from sound of it, is too infrequently considered at the present time by eye care specialists.  I would rather they had explored the topical aspect in greater depth and just given a nod to the systemic necessity, since the latter, while still a sadly common problem due to malnutrition, is not at all obscure.  

3) Twice the authors bring up their own studies of combined vitamin A, vitamin D and Omega 3 drops (pages 4 and 7).  While in a collective sense these investigations are certainly relevant to the discussion, i) since they tested all three in combination, they were in both circumstances logically forced to conclude "...the efficacy of the single components of the eye drop could not be ruled out."  Why didn't they attempt a subgroup evaluation to answer the inevitable questions, which would have been both informative and more definitive?  As presented, in each case (for vitamin A and vitamin D) these findings do nothing to illuminate what either compound may achieve topically;  ii) I'm well versed in the vitamin D literature particularly, and citing their own work while omitting considerable other pre-clinical and clinical efforts without explanation (there have been three US patents issued for topical treatment of dry eye using vitamin D preparations, for instance) left an impression---however hopefully unintended--that the authors were focused upon additionally promoting their own work in some sense.  The conflict of interest statement for this paper declared "no conflict," supporting the cited combination vitamin A, vitamin D and Omega 3 drops were not, presumably, representative of a product in development, but it didn't leave me with the sense of having received an unbiased and comprehensive review of the subject.  Again, I want to give the authors the benefit of the doubt as they are tackling an intricate, complicated subject, but perhaps this reflects the danger in attempting such an ambitious review.  

4) Lines 296 to 302 (page 8) the authors seem to be advancing the idea that "topical vitamin supplementation has the advantage...," again hinting at some interest in promoting this idea (?).  They offer that dry eye disease patients might better adhere to such an approach, too, rather than taking supplements systemically with associated issues of polytherapy and cost, yet do not consider that i) vitamin deficiency manifesting as ocular disease very likely reflects general systemic deficiency, with potentially  considerable--perhaps occult or sub-clinical--systemic disease requiring systemic treatment;  and ii) topical treatments are in many cases more expensive, particularly where they can qualify for patent protection under "vehicle" status.  These statements struck me as out of place in a review article and in addition unsubstantiated.  Once again, I found myself questioning the fundamental purpose of this paper.  I regret if that is unfair, but the authors deserve to understand how their audience may perceive their work, especially should it inspire adverse misinterpretation.
  All told, I felt I learned something novel (to me) and worthwhile from this paper in regard to the possible benefit of topical vitamin A therapy, but otherwise it left me with more questions than insights.  As an effective addition to the literature, I would prefer to see it more focused/refined or more overtly comprehensive and illuminating in the process.    

Author Response

Reviewer 2

I find the topic very interesting, so in that sense am positively inclined to see more work along these lines published, but that said while I keep trying to like this paper and it falls short still.  Trying to understand why I react that way to it, there seem to be three things:  

1) The authors survey 4230 papers and state they arrive at 254 providing pre-clinical and/or clinical evidence of efficacy, yet "in the end 55 studies were included in this review," which is approximately one out of five of those stipulated to meet their gross screening criteria.    No mention is made as to why the other four out of five were rejected.  Having explored these questions in some depth over 15 years myself, there are many studies of which I am aware which seem to have been dismissed.  Perhaps the authors had good reason to reject them, but we are given no clue as to why.  It is a very broad field of inquiry and I can't fault them for wanting to restrict their analysis to some digestible degree, but I would much rather they restrict their analysis to, say "topical vitamin impact" and offer more comprehensive exploration of their review criteria and findings than try to encompass the entire concept and leave me wondering why some studies were included and others not.

We are grateful to the Reviewer for the words of appreciation on this paper. In this revision we attempted to ameliorate it at our best, following Reviewers’ comments.

As recommended, we clarified the reasons for study exclusion.

2) Consideration of clinical studies on oral vitamin A, while perhaps needed to address the stated "systemic and topical" aspect of this paper, were not something which struck me as helpful (novel) contribution to the knowledge base.  Every ophthalmologist should receive textbook instruction regarding Vitamin A deficiency and xerophthalmia--nearly 100 years under study--and while perhaps the authors felt this needed due attention given the breadth of their topic, it was distracting to me. The discussion of topical supplementation was much more interesting and from sound of it, is too infrequently considered at the present time by eye care specialists. I would rather they had explored the topical aspect in greater depth and just given a nod to the systemic necessity, since the latter, while still a sadly common problem due to malnutrition, is not at all obscure.  

The Reviewer is right: the section on systemic Vitamin A supplementation in not novel at all as these concepts can be found in any textbook. We therefore tried to make this part as succinct as possible.

Yet, we decided to not to remove it for several reasons:

  1. to preserve the “systemic and topical” aspect of the paper, as correctly pointed out by the Reviewer him/herself. Excluding systemic supplementation would have removed the relevant info also from Vitamin D, leaving overall a poorly interesting paper
  2. the mechanisms for systemic supplementation are very useful to understand the topical effects
  3. two on three Reviewers asked more details on systemic Vit A supplementation!

We also tried to improve the discussion on topical supplementation.

3) Twice the authors bring up their own studies of combined vitamin A, vitamin D and Omega 3 drops (pages 4 and 7).  While in a collective sense these investigations are certainly relevant to the discussion, i) since they tested all three in combination, they were in both circumstances logically forced to conclude "...the efficacy of the single components of the eye drop could not be ruled out."  Why didn't they attempt a subgroup evaluation to answer the inevitable questions, which would have been both informative and more definitive? As presented, in each case (for vitamin A and vitamin D) these findings do nothing to illuminate what either compound may achieve topically; 

As also recommended by Reviewer 3, we merged this to Section 3.4

  1. ii) I'm well versed in the vitamin D literature particularly, and citing their own work while omitting considerable other pre-clinical and clinical efforts without explanation (there have been three US patents issued for topical treatment of dry eye using vitamin D preparations, for instance) left an impression---however hopefully unintended--that the authors were focused upon additionally promoting their own work in some sense. The conflict of interest statement for this paper declared "no conflict," supporting the cited combination vitamin A, vitamin D and Omega 3 drops were not, presumably, representative of a product in development, but it didn't leave me with the sense of having received an unbiased and comprehensive review of the subject. Again, I want to give the authors the benefit of the doubt as they are tackling an intricate, complicated subject, but perhaps this reflects the danger in attempting such an ambitious review.  

The preclinical evidence on vitamin D is very large; we tried to summarize it at our best and we are sorry if, for sake of brevity, we excluded some relevant paper. In this version we expanded this section, even though the main concepts on the efficacy are unchanged.

Concerning clinical efficacy of topical vitamin D3, we ran again pubmed database with “Cholecalciferol dry eye disease” or “vitamin D dry eye disease” or “Cholecalciferol ocular surface disease” or “vitamin D ocular surface disease”. No previously undetected studies were found (a case series on 6 patients was previously not included because of low number and lack of control, here we added it on discussion), so I’m sorry to say that the “clinical efforts” stated by the Reviewer cannot be mentioned if unavailable on Pubmed.

The product with Vitamin A, D and omega 3 is already clinically available; we do not have any conflict of interest on it but of course we are happy to cite the papers for which we previously worked hardly. Our feeling is that we are not promoting any product, but just showing clinical data on, as correctly said, an intricate subject.

In any case, we are more than happy to modify this section following specific indications by the Reviewer.

4) Lines 296 to 302 (page 8) the authors seem to be advancing the idea that "topical vitamin supplementation has the advantage...," again hinting at some interest in promoting this idea (?). 

They offer that dry eye disease patients might better adhere to such an approach, too, rather than taking supplements systemically with associated issues of polytherapy and cost, yet do not consider that i) vitamin deficiency manifesting as ocular disease very likely reflects general systemic deficiency, with potentially  considerable--perhaps occult or sub-clinical--systemic disease requiring systemic treatment; and ii) topical treatments are in many cases more expensive, particularly where they can qualify for patent protection under "vehicle" status.  These statements struck me as out of place in a review article and in addition unsubstantiated.  Once again, I found myself questioning the fundamental purpose of this paper.  I regret if that is unfair, but the authors deserve to understand how their audience may perceive their work, especially should it inspire adverse misinterpretation.

All told, I felt I learned something novel (to me) and worthwhile from this paper in regard to the possible benefit of topical vitamin A therapy, but otherwise it left me with more questions than insights.  As an effective addition to the literature, I would prefer to see it more focused/refined or more overtly comprehensive and illuminating in the process.    

We are sorry to read that the Reviewer had the impression that we attempted to use this review to drive the reader to prefer the use of topical vitamins. This is, honestly, not in the scope of our work. Yet the Reviewer is right in warning us on possible audience misinterpretation.

Actually, in this part of the Discussion we are just using (as commonly done) the classical thesis and antithesis process of any discussion.

The paragraph mentioned (lines 296-302) is the thesis: we are summarizing the possible advantages of topical administration (if any). Then the following paragraph is the antithesis: we refute the thesis and clearly state that only vitamin A has sufficient evidence of efficacy for topical use.

Perhaps you could be more convinced of our impartiality by reading again Abstract and Conclusions (both unmodified).

We are strongly convinced that systemic integration is ideal if there is a systemic deficiency - this has been made even clearer in this version. Yet, it is also our clinical impression that in local diseases, ie glaucoma patients, topical integration might be preferable, as well as for patients receiving multiple systemic treatments.

About cost, at least in Italy the cost of tear substitutes with or without vitamins or other microelements is nearly identical. Yet, we acknowledge that the Reviewer comment could be right outside our country.

Overall, considering the possible risk of audience misinterpretation, we moderated with attention the whole discussion, and we hope that the Reviewer will be satisfied by the changes.

A brief final comment on the ultimate scope of our review: following the clinical experience with lubricating eyedrops + vitamins, we had higher than expected results in some patients, a fact that lead us to study the literature on vitamin and ocular surface. Talking with colleagues we also understood how this topic is overall ignored. That’s why we starting writing it.   

Reviewer 3 Report

In this review, the authors reviewed 55 studies in this review. Authors described the results according to the vitamin that was used in those studies, and carefully evaluated the results depending on the sufficient information. Although the authors did a good job reviewing the related manuscripts, there are some suggestions regarding the arrangement of the contents for authors to consider:

  1. Both there are enough data to describe the effects of vitamin A and vitamin D on treating OSD or/and DED. However, the results of vitamin C and vitamin E are not too strong. Authors may re-group the results into 3.1 vitamin A; 3.2 vitamin B; 3.3 vitamin D; 3.4 vitamin C, E, or combined vitamins.
  2. In the column “author, year” of Table 1, authors may add the information of citation number for readers to connect the table and the contents. For example, “Babamohamadi, 2018” can be “Babamohamadi, 2018 [32]”.
  3. Authors repeatedly mentioned their own publications in section 3.1.3 and 3.4.3. However, these studies investigated the efficacy and safety of VisuEvo that contained vitamin A, vitamin E3, and Omega 3. It is difficult to evaluate whether the effect was resulted from the combination of three components or from single component. It might be better to describe these results if authors re-group the contents as suggested in point 1.

Author Response

Reviewer 3

In this review, the authors reviewed 55 studies in this review. Authors described the results according to the vitamin that was used in those studies, and carefully evaluated the results depending on the sufficient information. Although the authors did a good job reviewing the related manuscripts, there are some suggestions regarding the arrangement of the contents for authors to consider:

  1. Both there are enough data to describe the effects of vitamin A and vitamin D on treating OSD or/and DED. However, the results of vitamin C and vitamin E are not too strong. Authors may re-group the results into 3.1 vitamin A; 3.2 vitamin B; 3.3 vitamin D; 3.4 vitamin C, E, or combined vitamins.

We are thankful to Reviewer 3 for the words of appreciation on our paper. We modified the structure as recommended.

  1. In the column “author, year” of Table 1, authors may add the information of citation number for readers to connect the table and the contents. For example, “Babamohamadi, 2018” can be “Babamohamadi, 2018 [32]”.

Done as recommended

  1. Authors repeatedly mentioned their own publications in section 3.1.3 and 3.4.3. However, these studies investigated the efficacy and safety of VisuEvo that contained vitamin A, vitamin E3, and Omega 3. It is difficult to evaluate whether the effect was resulted from the combination of three components or from single component. It might be better to describe these results if authors re-group the contents as suggested in point 1.

Done as recommended

Round 2

Reviewer 1 Report

  1. Figure 1 describes MUC16, MUC 4, MUC 1. Please add the full term of those abbreviation in the figure legands
  2. Figure 1 describes MUC 1, MUC 4, MUC 1, are they the same as

mucin 1 in the 3.1.1. paragraph ? please make the term the same across the whole manuscript.

In addition, please add more description of MUC16  in the 3.1.1 paragraph.

  1. Line 75: vitamin A, Line 77: Vitamin A.

Vitamin A? vitamin A? please make the term the same across the manuscript. There are lots of these errors across the manuscript.

Meibomian gland? or meibomian gland?

Cycloporine? Cyclosporine?

Line 334 : countries? Countries?

Topical treatment ? or local treatment?

4 Please add the full term of abbreviation IL-2 (Line 165) , OSDI(Line 176) 95% IC? Or 95% CI (Line 226)

  1. Table 1: citology? cytology?
  2. Line 154, it might be better to describe more about dexpanthenol. Readers will be confused about vitamin B and dexpanthenol 

Author Response

Reviewer 1

  1. Figure 1 describes MUC16, MUC 4, MUC 1. Please add the full term of those abbreviation in the figure legands

Abbreviations have been removed also from Figure

  1. Figure 1 describes MUC 1, MUC 4, MUC 1, are they the same as

mucin 1 in the 3.1.1. paragraph ? please make the term the same across the whole manuscript.

Done as recommended

In addition, please add more description of MUC16  in the 3.1.1 paragraph.

Done as recommended

  1. Line 75: vitamin A, Line 77: Vitamin A.

Vitamin A? vitamin A? please make the term the same across the manuscript. There are lots of these errors across the manuscript.

Meibomian gland? or meibomian gland?

Cycloporine? Cyclosporine?

Line 334 : countries? Countries?

Topical treatment ? or local treatment?

4 Please add the full term of abbreviation IL-2 (Line 165) , OSDI(Line 176) 95% IC? Or 95% CI (Line 226)

  1. Table 1: citology? cytology?

These errors have been corrected 

  1. Line 154, it might be better to describe more about dexpanthenol. Readers will be confused about vitamin B and dexpanthenol 

We clarified that it is a provitamin and an alcohol analog of vitamin B5 (pantothenic acid).

Reviewer 2 Report

1) A few minor spelling issues left (example, Section 3.1 revision "From the data available on literature..."

2) Perhaps what has most led me to feel the paper fell short of promise, in retrospect, is this line from the introduction:

Recently many lubricating eyedrops have been enriched with vitamins as some studies showed them  to help promoting OS homeostasis and controlling inflammation. 

I am unaware of "many" lubricating eyedrops being enriched with vitamins, though perhaps there are more available internationally than in our area.  Then the following statement "some studies showed them to help promoting OS homeostasis and controlling inflammation" left me expecting one or more references to be provided for those studies.   Being in the introduction, this may have been intended just as a lead-in to the sections to follow, but it seemed to suggest there were already products in production resulting from evidence we were to be shown at some point, which never quite occurred.  This proved more of a disappointment than a fault, but I think it did lend me to be more critical of the remainder of the paper as I sought/scoured it for support for the claim.

3) I wish in sorting the literature to review, perhaps meta-analysis could have been done for some number of the case report papers with fewer than 10 participants.  There may have been useful information or findings revealed about topical vitamin C or D in isolation, for instance, which otherwise yielded essentially nothing in the selected studies.  Hard to imagine nothing in the Pub Med covered world literature about either, so I was left wondering if there may be smaller studies or reports which were rejected due to their size.

Author Response

Reviewer 2

1) A few minor spelling issues left (example, Section 3.1 revision "From the data available on literature..."

Corrected as recommended

2) Perhaps what has most led me to feel the paper fell short of promise, in retrospect, is this line from the introduction:

Recently many lubricating eyedrops have been enriched with vitamins as some studies showed them  to help promoting OS homeostasis and controlling inflammation. 

I am unaware of "many" lubricating eyedrops being enriched with vitamins, though perhaps there are more available internationally than in our area.  Then the following statement "some studies showed them to help promoting OS homeostasis and controlling inflammation" left me expecting one or more references to be provided for those studies.   Being in the introduction, this may have been intended just as a lead-in to the sections to follow, but it seemed to suggest there were already products in production resulting from evidence we were to be shown at some point, which never quite occurred.  This proved more of a disappointment than a fault, but I think it did lend me to be more critical of the remainder of the paper as I sought/scoured it for support for the claim.

We understand the Reviewer viewpoint. In this revision, we mitigated again that sentence of Introduction. We removed the claim to studies, and included other compounds (aminoacids and sugars) to make the view of this sentence even larger.

3) I wish in sorting the literature to review, perhaps meta-analysis could have been done for some number of the case report papers with fewer than 10 participants.  There may have been useful information or findings revealed about topical vitamin C or D in isolation, for instance, which otherwise yielded essentially nothing in the selected studies.  Hard to imagine nothing in the Pub Med covered world literature about either, so I was left wondering if there may be smaller studies or reports which were rejected due to their size.

The choice of removing papers with less than 10 patients is arbitrary; it was done to exclude small case series which, in most cases, do not have the proper structure of a clinical study but are more focused on the observation on single cases. In general, poor internal validity is the reason for exclusion from meta-analysis.